# The Effects of Longer Use of Teriparatide on Clinical and Radiographic Outcomes after Spinal Fusion in Geriatric Patients

**DOI:** 10.3390/medicina60060946

**Published:** 2024-06-06

**Authors:** Young-Hoon Kim, Kee-Yong Ha, Hyun W. Bae, Hyung-Youl Park, Young-Il Ko, Myung-Sup Ko, Sang-Il Kim

**Affiliations:** 1Department of Orthopedic Surgery, Seoul St. Mary’s Hospital, College of Medicine, The Catholic University of Korea, 222, Banpo-daero, Seocho-gu, Seoul 06591, Republic of Korea; 2Department of Orthopedic Surgery, Cedars-Sinai Medical Center, Los Angeles, CA 90048, USA; 3Department of Orthopedic Surgery, Eunpyeong St. Mary’s Hospital, College of Medicine, The Catholic University of Korea, Seoul 06591, Republic of Korea

**Keywords:** geriatric, lumbar fusion, teriparatide, treatment duration, osteoporosis

## Abstract

*Background*: Teriparatide is an anabolic agent for osteoporosis and is believed to improve the bone healing process. Previous studies showed that teriparatide could enhance not only fracture healing but also spine fusion. It has been reported that use of teriparatide could promote the spine fusion process and decrease mechanical complications. However, there was no consensus regarding optimal treatment duration. The purpose of this study was to compare surgical outcomes between short-duration and long-duration teriparatide treatment after lumbar fusion surgery in elderly patients. *Materials and Methods*: All consecutive patients older than 60 years who underwent 1-level lumbar fusion surgery for degenerative diseases between January 2015 and December 2019 were retrospectively reviewed. Based on the duration of teriparatide treatment (daily subcutaneous injection of 20 µg teriparatide), patients were subdivided into two groups: a short-duration (SD) group (<6 months) and a long-duration (LD) group (≥6 months). Mechanical complications, such as screw loosening, cage subsidence, and adjacent vertebral fractures, were investigated. Postoperative 1-year union rate was also evaluated on computed tomography. Clinical outcomes were recorded using visual analog scale (VAS) and Oswestry Disability Index (ODI). Between-group differences for these radiographic and clinical outcomes were analyzed. *Results*: Ninety-one patients were reviewed in this study, including sixty patients in the SD group and thirty-one patients in the LD group. Their mean age was 72.3 ± 6.2 years, and 79 patients were female. Mean T-score was −3.3 ± 0.8. Cage subsidence (6.7% vs. 3.2%), screw loosening (28.3% vs. 35.5%), and adjacent vertebral fracture (6.7% vs. 9.7%) were not significantly different between the SD and LD groups. Union rate at 1-year postoperative was 65.0% in the SD group and 87.1% in the LD group (*p* = 0.028). Both groups showed improvement in VAS and ODI after surgery. However, the differences of VAS from preoperative to 6 months and 1 year postoperative were significantly higher in the LD group. *Conclusions*: Longer teriparatide treatment after lumbar fusion surgery resulted in a higher union rate at 1-year postoperative than the shorter treatment. Also, it could be more beneficial for clinical outcomes.

## 1. Introduction

With the global aging, geriatric patients are rapidly increasing worldwide in almost all medical fields, and patients with degenerative spine diseases are no exception. For example, a recent Finnish nationwide data study showed that lumbar fusion surgery increased from 1997 to 2018, particularly increasing the most in women aged over 75 years, with a 4-fold increase [1]. Although fusion surgery usually yields favorable outcomes, one of the inherent complications is nonunion. Previous reports showed that the incidence of nonunion ranges from 5% to 35% [2]. Nonunion is also estimated to be one of the main causes of revision surgeries [3]. 

Osteoporosis, an age-related metabolic bone disease, is also increasing in the aging population. Poor bone quality is well known to increase the risk of several postoperative complications, including nonunion, fracture, and instrumentation failure [4,5,6]. It was reported that revision surgeries were more common in osteoporotic patients than in patients without osteoporosis [7]. Thus, it is becoming more and more important preoperatively to perform bone mineral density (BMD) and to optimize bone quality, if needed. Parathyroid hormone (PTH) is produced by the parathyroid and is involved in calcium and phosphate homeostasis. Teriparatide, a recombinant human PTH analog, is a powerful agent commonly used to treat osteoporosis since receiving FDA approval in 2002. It induces the maturation and differentiation of osteoblast precursors, stimulates the pre-existing osteoblasts for new bone formation, and prevents osteoblasts/osteoclast apoptosis [8]. Because teriparatide enhances bone formation, which is greater in bones with increased bone metabolism, cancellous bones show greater increases in bone density compared to cortical bones [9]. Furthermore, it is thought to be able to enhance the bone healing process due to its anabolic effect by stimulating osteoblasts [10]. Multiple studies have demonstrated that teriparatide could improve the fracture healing process [11,12,13]. Although there is a clear risk of bias, some studies show positive results even in the healing of spinal fractures [14,15]. The bone healing process after spinal fusion surgery is very similar to the fracture healing process. Therefore, it has been speculated that teriparatide may also improve this spinal fusion process, and several trials have been performed in order to test this theory [16,17,18,19,20]. Ebata et al., in their randomized controlled study, reported that fusion was significantly higher in the teriparatide arm than in the control arm at 6-month follow-up [21].

Although teriparatide is now believed to be able to improve surgical outcomes and reduce complications after spinal fusion surgery, there is a paucity of evidence showing the relationship between surgical outcomes and its treatment duration. Thus, the purpose of this study was to compare the surgical outcomes between short-duration and long-duration teriparatide treatment after lumbar interbody fusion surgery in elderly patients.

## 2. Materials and Methods

### 2.1. Patients

This study was approved by the Institutional Review Board before its initiation. Patients who underwent 1-level interbody fusion surgery for degenerative lumbar diseases in a single tertiary institute between 2015 and 2019 were retrospectively analyzed. Inclusion criteria were age of 65 years or more, the use of teriparatide for at least one month after fusion surgery, and at least 1-year follow-up. Exclusion criteria were lateral access surgery, any surgical history at the index level, preoperative use of teriparatide, contraindications of teriparatide (hypersensitivity to teriparatide, pre-existing hypercalcemia, hyperparathyroidism, known bone metastasis, prior radiotherapy involving bone, and Paget’s disease of bone), use of bone morphogenetic protein, concomitant use of anti-resorptive agents, and insufficient data. We investigated the medical comorbidities and prescribed medicines of all patients. To quantitatively compare comorbidities, we used the modified Charlson Comorbidity Index (mCCI). All surgeries were performed by two experienced orthopedic spine surgeons using the same manner: midline incision, and posterior or transforaminal interbody fusion with titanium cages, local autograft, and demineralized bone matrix followed by posterior instrumentation using pedicle screws and rod. If patients were not contraindicated, teriparatide treatment started within 7 days after surgery. Although we recommended that all patients should be treated for as long as possible, the individual treatment duration was decided totally based on the patient’s own circumstances. After discontinuation of teriparatide, antiresorptive agents were administered only in patients with osteoporosis. Patients were subdivided into two groups based on treatment duration: a short-duration (SD) group (<6 months) and a long-duration (LD) group (≥6 months).

### 2.2. Data

Baseline demographic data, including age, sex, and body mass index (BMI), were recorded. Bone mineral density (BMD) was performed preoperatively in all patients using dual-energy X-ray absorptiometry (DXA). Osteoporosis was diagnosed when the T-score was less than −2.5 on DXA. For patient-reported outcomes (PROs), Oswestry Disability Index (ODI) and visual analog scale (VAS) for back pain were reviewed. ODI is one of the principal patient-reported outcomes that measures a functional disability for spinal disorders [22]. This self-completed questionnaire consists of 10 questions which have 6 statements that are scored from 0 to 5. The scores for each question are summed and multiplied by two. Therefore, ODI ranges from 0 to 100 and is displayed as %, where 0% indicates no disability and 100% indicates the most severe disability. For the assessment of fusion status and mechanical complications, computed tomography (CT) scan with 1-mm slice was conducted. Fusion was assessed by CT scan at 1-year postoperative based on the Brantigan, Steffee, and Fraser (BSF) criteria: grade 1, pseudoarthrosis; grade 2, locked pseudoarthrosis; and grade 3, fusion [23]. In our study, grades 1 and 2 were considered as nonunion. We investigated mechanical complications such as screw loosening, cage subsidence, and adjacent vertebral fractures by plain radiographs and CT scan. Screw loosening was considered present when a radiolucent hollow of more than 1 mm and a radiopaque rim around the screw were identified on plain radiographs or CT scan. Cage subsidence was considered when more than 2 mm migration of the cage into the adjacent vertebral body was found on lateral radiographs or CT scan.

### 2.3. Statistical Analysis

Statistical analysis was performed using SPSS software version 24.0.0 (SPSS Inc., Chicago, IL, USA). Independent *t*-test and chi-square test were used for continuous and categorial data between two groups, respectively. In this study, a *p*-value < 0.05 was considered statistically significant.

## 3. Results

A total of 91 patients were included in this study. Their mean age was 72.3 ± 6.2 years, and 79 patients (86.8%) were female. Mean BMI, mean T-score, and mean mCCI were 24.2 ± 3.8, −3.3 ± 0.8, and 1.1 ± 1.2, respectively. Among the patients included in this study, 60 patients used teriparatide for less than 6 months (SD group) and 31 patients used it for more than 6 months (LD group). Mean duration of teriparatide treatment was 7.8 ± 1.8 months in the LD group and 3.0 ± 1.1 months in the SD groups. Comparisons of baseline demographic data between the two groups are shown in Table 1. There were no significant differences in age, sex, BMI, BMD, and mCCI between the two groups.

### Radiographic and Clinical Outcomes

Details about radiographic complications are shown in Table 2. Nonunion at 1 year was identified in 21 and 4 patients of the SD and LD groups, respectively, with a higher incidence in the SD group (35.0% vs. 12.9%, *p* = 0.028). However, incidences of screw loosening (28.3% for SD vs. 35.5% for LD, *p* = 0.484), cage subsidence (6.7% for SD vs. 3.2% for LD, *p* = 0.658), and adjacent vertebral fracture (6.7% for SD vs. 9.7% for LD, *p* = 0.609) were not significantly different between the two groups. There were two patients who required reoperation during the study period: one for surgical site infection and another for cage migration.

PROs at each time point are shown in Table 3. Preoperative ODI and VAS for back pain were not significantly different between the two groups and improved after surgery in both groups. Changes of these two PROs were not significantly different at 3 months postoperative. However, they showed significant differences at 6 months after surgery. Differences of VAS from preoperative to 6 months postoperative were −2.5 ± 2.6 and −3.9 ± 2.8 in the SD and LD groups, respectively. They were significantly higher in the LD group (*p* = 0.033). Differences of ODI from preoperative to 6 months postoperative were −14.8 ± 23.6 and −24.5 ± 21.0 in the SD and LD groups, respectively, showing no significant difference between the two groups (*p* = 0.095). At 1-year postoperative, the difference in VAS was significantly higher in the LD group (−3.0 ± 3.6 in SD vs. −4.6 ± 2.2 in LD group, *p* = 0.023). Changes of ODI from preoperative to 1-year postoperative were −19.9 ± 26.0 and −31.5 ± 17.7 in the SD and LD groups, respectively (*p* = 0.055).

## 4. Discussion

Intersegmental spinal fusion requires an imbalance between bone formation and resorption towards bone formation at the site of bone graft. Therefore, an increase in osteoblastic activity, a decrease in osteoclastic activity, or both can promote the fusion process. Intermittent PTH administration can increase osteoblast activity, ultimately enhancing bone mass by increasing cancellous bone volume [24,25]. Since the introduction of teriparatide as a treatment for osteoporosis in the early 2000s, its anabolic effect on osteoblast activation has led to several studies analyzing the impact of teriparatide in spinal fusion surgery [16,17,18,19,20,26]. Ohtori et al. conducted the first clinical trial to analyze the effect of teriparatide after instrumented lumbar posterolateral fusion [16]. Teriparatide was administered for 2 months before surgery and for 8 months after surgery. They showed that teriparatide could increase the fusion rate and decrease the time to fusion compared to bisphosphonate at 1-year follow-up. However, no significant differences in pain and disability scores were observed in their study. Cho et al., in their prospective study of 47 patients undergoing posterior lumbar interbody fusion (PLIF), administered teriparatide for 3-month cycles alternating with 3-month periods of oral daily bisphosphonate (BP) for 12 months [17]. They reported that there were no significant differences in the overall fusion rate or clinical outcome, although the teriparatide group showed an earlier fusion than the BP group. Our results showed that the fusion rate at 1 year was significantly higher in the LD group than in the SD group, while the complication rate was similar between the two groups. A recent systematic review and meta-analysis reported that teriparatide could improve the fusion rate and decrease the risk of screw loosening compared to the control in which teriparatide had not been used [27]. In contrast, another meta-analysis showed that teriparatide has no effect on fusion rate, although it might be associated with lower screw loosening [28]. Summarizing previous studies, teriparatide may improve the fusion rate or reduce postoperative complications after spine fusion surgery.

The heterogeneity of osteoporosis medication intervention in previous clinical studies has been commonly identified in most meta-analyses [27,28,29]. Xiong et al. compared 36 patients who received daily teriparatide with 41 patients who received zoledronate for 2 years after PLIF and found that the union rate was higher in the teriparatide group at 6 months but similar at 2 years [19]. The study by Wang et al. included 115 patients who underwent transforaminal lumber interbody fusion and administered teriparatide for at least 6 months after surgery. They found that the union rate in the teriparatide group was not superior to that in the group of patients with a single dose of zoledronate at 6 months, but greater at 1 year in the teriparatide group [30]. However, Jespersen et al. reported that 90-day administration of teriparatide did not increase fusion volume or improve the quality of the fusion mass compared to placebo after non-instrumented spinal fusion [18].

Although teriparatide may improve surgical outcomes, the optimal duration of teriparatide use remains unclear. Few studies have assessed the effect of teriparatide treatment duration on fusion. To the best of our knowledge, the clinical study by Ohtori et al. is the only one that has analyzed differences in surgical outcomes after short- and long-term use of teriparatide after spine fusion [31]. They found that fusion was more efficient regarding the rate (90% vs. 83% at 15 months) and mean duration (7.7 vs. 9.3 months) in the long-term use group than in the short-term use group, although they did not assess clinical outcomes. In our study, VAS and ODI improved consistently in both groups after surgery. However, the amount of improvement in VAS was significantly greater in the LD group than in the SD group at 6 months and 1 year after surgery. Although the amount of improvement in ODI was not significantly different between the two groups, the LD group showed greater improvement with proximity to statistical significance at 1 year. It remains unclear whether teriparatide treatment could affect clinical outcomes after spine fusion surgery. Some studies have shown that teriparatide use did not have a favorable effect on ODI compared to control [16,17,21,30], while others have shown significantly greater ODI improvement with teriparatide use [19,20]. Two clinical studies by Xiong et al. and Seki et al. reported better clinical results in the teriparatide group administered for a long duration (2 years) [19,20]. We believed that longer use of teriparatide might be more beneficial in terms of clinical as well as radiographic outcomes.

This study has several limitations. First, this study had a small sample size and a retrospective design known to have inherent selection bias. Second, the follow-up period and treatment period were relatively short. We could not find if longer use (>1 year) would be more favorable. Third, patients with antiresorptive agents were not included. Bisphosphonate also could have a positive effect on fusion rate [28]. Fourth, the cost effectiveness of teriparatide was not evaluated. Although teriparatide was used not only for fusion but also for BMD increase in our cohort, its cost effectiveness should be considered clinically. Teriparatide may also interact with some drugs such as digoxin, furosemide, hydrochlorothiazide, etc. [32]. These drugs can alter calcium levels, but we routinely check calcium levels before the administration and excluded patients with hypercalcemia. Although geriatric patients usually take several kinds of medications for their chronic endocrinological, cardiological, and neurological diseases, this study did not analyze the medications of our patients. However, to the best of our knowledge, no drugs are known to affect the anabolic action of teriparatide. Therefore, it was believed that the drugs taken by patients had no or minimal effects on the outcomes. Finally, we used an arbitrary time point (6 months) to divide patients. We could not find out how long teriparatide treatment is needed to improve postoperative outcomes.

## 5. Conclusions

Longer teriparatide treatment may improve the fusion rate in elderly patients who undergo spine fusion surgery. Additionally, longer teriparatide treatment periods might be more beneficial for clinical outcomes than shorter treatment periods. Further large-scale studies are needed to find the optimal duration of teriparatide treatment in order to enhance surgical outcomes in geriatric patients.

## Figures and Tables

**Table 1 medicina-60-00946-t001:** Baseline demographic data of short- and long-duration groups.

	Short Duration (*n* = 60)	Long Duration (*n* = 31)	*p*
Age	72.0 ± 6.9	72.9 ± 4.8	0.562 *
Sex (Female)	53 (88.3%)	26 (83.9%)	0.551 #
BMI	23.8 ± 3.4	24.9 ± 4.3	0.268 *
BMD	−3.3 ± 0.8	−3.2 ± 0.9	0.612 *
Osteoporosis diagnosis	49 (81.7%)	25 (80.6%)	0.906 #
mCCI	1.1 ± 1.2	1.0 ± 1.2	0.697 *

* Independent *t*-test and # chi-square test were performed.

**Table 2 medicina-60-00946-t002:** Comparison of radiographic outcomes between short- and long-duration groups.

	Short Duration (*n* = 60)	Long Duration (*n* = 31)	*p*
Nonunion at 1 year (CT)	21 (35.0%)	4 (12.9%)	**0.028**
Screw loosening	17 (28.3%)	11 (35.5%)	0.484
Cage subsidence	4 (6.7%)	1 (3.2%)	0.658
Adjacent vertebral fracture	4 (6.7%)	3 (9.7%)	0.609

Chi-square test was performed. A bold value indicates statistical significance.

**Table 3 medicina-60-00946-t003:** Comparison of clinical outcomes between short- and long-duration groups.

	Short Duration (*n* = 60)	Long Duration (*n* = 31)	*p*
VAS			
Preop	7.3 ± 2.6	6.9 ± 2.8	0.522
3 months	4.7 ± 2.4	4.2 ± 2.6	0.448
6 months	4.8 ± 2.1	3.1 ± 2.0	**0.002**
1 year	4.4 ± 2.7	2.6 ± 2.1	**0.008**
Δ3 months-preop	−2.6 ± 3.2	−2.7 ± 3.2	0.951
Δ6 months-preop	−2.5 ± 2.6	−3.9 ± 2.8	**0.033**
Δ1 year-preop	−3.0 ± 3.6	−4.6 ± 2.2	**0.023**
ODI (%)			
Preop	62.7 ± 17.6	57.0 ± 13.8	0.177
3 months	57.6 ± 16.8	56.2 ± 17.2	0.750
6 months	47.9 ± 17.0	34.0 ± 18.5	**0.003**
1 year	44.6 ± 18.6	27.9 ± 16.4	**0.001**
Δ3 months-preop	−5.1 ± 20.9	−0.8 ± 17.2	0.392
Δ6 months-preop	−14.8 ± 23.6	−24.5 ± 21.0	0.095
Δ1 year-preop	−19.9 ± 26.0	−31.5 ± 17.7	0.055

Δ means the difference between values at each time point. Independent *t*-test was performed. A bold value indicates statistical significance.

## Data Availability

The raw data supporting the conclusions of this article will be made available by the authors on request.

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
