# Peer review of "The Effects of Longer Use of Teriparatide on Clinical and Radiographic Outcomes after Spinal Fusion in Geriatric Patients"

_medicina, 2024, doi:10.3390/medicina60060946_

Round 1

Reviewer 1 Report

Comments and Suggestions for Authors

Dear Authors

minor revision is neccesary.

1. Please write much more about the Teriparatide. Even in abstract something in bracket.

2. Vas scale is known, but write more about ODI scale

3. The is the information about your Institution agreedment. Institutional Review Board: if the decision have a number, for example: 2/23, plase write it. If not plase write (not numbered)

4. in the table numer 3, the is the sign: Δ. Plase explain whot does it mean

5 Keywords: geriatric; lumbar fusion; teriparatide; outcome; treatment duration; osteoporosis

please remove: outcome, 

Reviewer 2 Report

Comments and Suggestions for Authors

I find a fundamental flaw in the article based on my research.

The article concerns phamatotherapy. The aim of the study is to find the optimal duration of teriparatide treatment to enhance surgical outcomes in geriatric patients.

There is no information in the article about other drugs taken, in particular cardiological, endocrinological and neurological drugs. None. There is no distinction of appropriate subgroups of the subjects, depending on the drugs used. This is a group of geriatric patients, so certainly some patients must take such medications. There is no information about this in the results. No response to this problem in the discussion. These problems were also not indicated in the methodology.

In my opinion, this is necessary when publishing articles bordering on pharmacology.

This also applies to the analysis of other diseases, apart from osteoporosis, suffered by patients from the study group.

This is a major defect in the article that requires improvement.
